# HDL Function and Size in Patients with On-Target LDL Plasma Levels and a First-Onset ACS

**DOI:** 10.3390/ijms24065391

**Published:** 2023-03-11

**Authors:** Alberto Cordero, Natàlia Muñoz-García, Teresa Padró, Gemma Vilahur, Vicente Bertomeu-González, David Escribano, Emilio Flores, Pilar Zuazola, Lina Badimon

**Affiliations:** 1Cardiology Department, Hospital Universitario de San Juan, 03550 Alicante, Spain; 2Unidad de Investigación en Cardiología, Fundación Para el Fomento de la Investigación Sanitaria y Biomédica de la Comunitat Valenciana (FISABIO), 46020 Valencia, Spain; 3Centro de Investigación Biomédica en Red de Enfermedades Cardiovasculares (CIBERCV), 28029 Madrid, Spain; 4Cardiovascular-Program ICCC, Institut d’Investigació Biomèdica Sant Pau (IIB SANT PAU), 08041 Barcelona, Spain; 5Departamento de Análisis Clínicos, Hospital Universitario de San Juan, 03550 Alicante, Spain; 6Cardiovascular Research Chair, Autonomous University of Barcelona, 08193 Barcelona, Spain

**Keywords:** ACS, NSTEMI, STEMI, lipoprotein functionality, HDL-C, LDL-C, cholesterol efflux

## Abstract

Patients admitted for acute coronary syndrome (ACS) usually have high cardiovascular risk scores with low levels of high-density lipoprotein cholesterol (HDL-C) and high low-density lipoprotein cholesterol (LDL-C) levels. Here, we investigated the role of lipoprotein functionality as well as particle number and size in patients with a first-onset ACS with on-target LDL-C levels. Ninety-seven patients with chest pain and first-onset ACS with LDL-C levels of 100 ± 4 mg/dL and non-HDL-C levels of 128 ± 4.0 mg/dL were included in the study. Patients were categorized as ACS and non-ACS after all diagnostic tests were performed (electrocardiogram, echocardiogram, troponin levels and angiography) on admission. HDL-C and LDL-C functionality and particle number/size by nuclear magnetic resonance (NMR) were blindly investigated. A group of matched healthy volunteers (*n* = 31) was included as a reference for these novel laboratory variables. LDL susceptibility to oxidation was higher and HDL-antioxidant capacity lower in the ACS patients than in the non-ACS individuals. ACS patients had lower HDL-C and Apolipoprotein A-I levels than non-ACS patients despite the same prevalence of classical cardiovascular risk factors. Cholesterol efflux potential was impaired only in the ACS patients. ACS-STEMI (Acute Coronary Syndrome—ST-segment-elevation myocardial infarction) patients, had a larger HDL particle diameter than non-ACS individuals (8.4 ± 0.02 vs. 8.3 ± 0.02 and, ANOVA test, *p* = 0.004). In conclusion, patients admitted for chest pain with a first-onset ACS and on-target lipid levels had impaired lipoprotein functionality and NMR measured larger HDL particles. This study shows the relevance of HDL functionality rather than HDL-C concentration in ACS patients.

## 1. Introduction

Coronary artery disease (CAD) is the major cause of mortality worldwide and is characterized by the chronic and initially silent development of atherosclerotic plaques in the coronary arteries [1,2]. Acute coronary syndromes (ACSs) are unstable and abrupt clinical manifestations of atherosclerosis, including a wide range of presentations such as unstable angina, non-ST-segment-elevation myocardial infarction (NSTEMI) and ST-segment-elevation myocardial infarction (STEMI) [3].

High levels of low-density lipoprotein cholesterol (LDL-C) is the leading effector for atherosclerosis development and, also, for recurrent cardiovascular events after an initial ACS [4]. Conversely, high-density lipoprotein cholesterol (HDL-C) is a strong independent predictor that inversely correlates with the risk of CAD and its thrombotic complications [5,6,7,8,9]. However, there are controversial results regarding HDL-C levels and CAD coming from Mendelian randomization studies [10] and pharmacological studies raising HDL-C levels [11]. Therefore, a new concept has arisen considering that cholesterol carried by HDL (HDL-C) does not reflect HDL functionality; in fact, it is the HDL micelle and not HDL-C that has shown different anti-atherogenic properties [12,13,14]. Moreover, traditional measures of cholesterol quantify the cholesterol and triglyceride content of lipoproteins in milligrams per decilitre and use the amount of cholesterol measured to assess risk. However, individuals can vary in their lipoprotein particle numbers and sizes, meaning that even though they might have equivalent cholesterol levels, they can vary in their risk for cardiovascular disease (CVD). Measuring particle number and size by nuclear magnetic resonance (NMR) spectroscopy could be a better read-out for CVD risk assessment [15,16].

Clinical registries have highlighted that a large percentage of patients admitted for an ACS have non-elevated LDL-C levels [8,9] and that atherosclerotic plaques can be detected even in patients with very low levels of LDL-C [17]. Therefore, HDL function and its interplay with other lipid and non-lipid molecules represent a challenge in ACS risk and onset [18,19].

Based on this evidence, we designed a real-world clinical study (as indicated in the Graphical Abstract) to investigate HDL/LDL functionality and lipoprotein particle number and size in patients with a first-onset ACS presentation having LDL-C and non-HDL-C with average levels of 100 ± 3.6 mg/dL and 128 ± 4.0 mg/dL, respectively, and intermediate cardiovascular risk scores.

## 2. Results

### 2.1. Patient Characteristics

Ninety-seven subjects (72 men, 25 women) with an average age of 64.9 ± 1.2 years who were initially recruited for the study were included in the final analysis. Categorization of the patients according to electrocardiogram pattern on admission (Table 1) found that (NSTEMI) patients were older (mean ± SEM; 70.3 ± 1.9) and more likely to have hypertension, diabetes mellitus (DM) and dyslipidaemia than STEMI patients. Conversely, STEMI patients were younger (mean ± SEM; 60.0 ± 1.7) and had higher smoking habits. No differences regarding body mass index (BMI) were reported between groups (*p* = 0.354). HDL-C and apolipoprotein A-I (ApoA-I) levels were significantly lower in (ACS) patients compared with non-ACS patients. Nevertheless, no differences were observed in LDL-C and non-HDL-C levels between groups. The STEMI group had lower left ventricular ejection fraction (LVEF) and higher cardiac damage (troponins) than the NSTEMI and non-ACS groups (Student’s *t*-test, *p* = 0.007 and *p* < 0.001, respectively) and more subsequent intervention for revascularization (χ2 test, *p* < 0.001) and coronary stents (χ2 test, *p* = 0.007). Interestingly, preadmission medication use was higher in the NSTEMI and non-ACS groups, the use of acetylsalicylic acid (ASA) (χ2 test, *p* = 0.044), angiotensin receptor blocker (χ2 test, *p* = 0.048), calcium antagonists (χ2 test, *p* = 0.022) and beta-blockers (χ2 test, *p* = 0.004) being significantly higher.

As shown in Appendix A, patients had an average of 100 ± 3.6 mg/dL of LDL-C at arrival, as well as 146 ± 8.8 mg/dL of triglycerides and 128 ± 4.0 mg/dL of non-HDL-C (on-target levels for primary prevention).

### 2.2. Lipoprotein Particle Number and Size

As shown in Table 2, ACS patients (especially STEMI) had a decreased number of small HDL particles (HDL-P) (ANOVA test, *p* < 0.001) compared with non-ACS individuals. Overall, ACS patients had a larger HDL particle diameter than non-ACS individuals (8.4 ± 0.02 vs. 8.3 ± 0.02 and, ANOVA test, *p* = 0.004). The triglyceride content (mg/dL) was similar in all lipoproteins.

Particle numbers of each lipoprotein class (expressed in percentage) are provided in Appendix A. ACS patients (especially STEMI) had a lower percentage of small very low-density but a higher percentage of medium small very low-density particles (VLDL-P) compared with non-ACS individuals (Student’s *t*-test for unpaired samples, *p* = 0.026 and *p* = 0.020, respectively). Furthermore, ACS patients (especially STEMI) had a lower percentage of small HDL particles (HDL-P) (ANOVA test, *p* = 0.004) but higher medium (ANOVA test, *p* = 0.004) and large HDL-P percentages (ANOVA test, *p* = 0.012) compared with non-ACS individuals.

As shown in Appendix A, the reference group of volunteers without cardiovascular disease had a lower proportion of pro-atherogenic LDL particles despite having higher levels of LDL-C, though these differences did not reach statistical significance. However, similarly to non-ACS individuals, the reference group had a smaller average particle diameter of HDL compared with ACS patients (Bonferroni post hoc test: reference vs. NSTEMI *p* = 0.004 and reference vs. STEMI, *p* < 0.001).

### 2.3. Assays for Lipoprotein Functionality

Although patients had on-target LDL-C levels (lower than the reference healthy group), LDL susceptibility to oxidation was higher in patients than in the reference individuals (*p* < 0.001, Figure 1A), presenting approximately a 10 min faster LDL oxidation (time to half maximum) even in the presence of medication. In addition, HDL antioxidant capacity and cholesterol efflux were also impaired in ACS patients compared with the reference group (Figure 1B,C; Student’s *t*-test, *p* = 0.002 and *p* = 0.038, respectively).

Differences in lipoprotein function are shown in Table 3. ACS patients exhibited a diminished capacity to promote cholesterol efflux with respect to the reference subjects (CEC (%):19.5 ± 0.6 vs. 22.3 ± 1.4; Student’s *t*-test, *p* = 0.041). By TRAP analysis (% of oxidized LDL inhibition), both non-ACS and ACS groups exhibited an impaired HDL-antioxidant capacity compared with the healthy population (ANOVA test, *p* = 0.005).

### 2.4. HDL Oxidation Inversely Correlates with Cholesterol Efflux Capacity (CEC)

In the reference group, induced HDL oxidation (determined by fluorometry) was inversely correlated with the CEC (Figure 2); interestingly, a similar relationship was found in non-ACS patients (Figure 2A and Appendix A). However, ACS-STEMI patients had a significantly lower slope and Y-intercept value than the reference group (*p* = 0.029 and *p* = 0.012, respectively) after inducing HDL oxidation (Figure 2B and Appendix A) (*p* = 0.003 and *p* < 0.001, respectively). In summary, there was a significant alteration in HDL function in ACS patients; in fact, in the STEMI group there was no relation to CEC. Both non-oxidized HDL and oxidized HDL was unable to promote CEC in STEMI patients.

We have carefully analysed the time of blood collection with respect to patient admission to observe changes in the lack of HDL function in STEMI patients (Appendix A). There was no time effect on the reduced HDL function regarding CEC and TRAP or in particle size distribution in samples collected just after admission or more than 24 h later. ACS patients that had percutaneous coronary intervention and whose blood was collected more than 24 h later had the lowest plasma HDL/ApoA-I levels. Therefore, HDL functionality was not altered during the acute phase in ACS patients.

## 3. Discussion

Our study in a cohort of real-world patients with non-elevated LDL-C levels and admitted for a first ACS, demonstrates the importance of taking into account HDL/LDL functionality and lipoprotein particle number/size in ACS patients as an improved read-out for CVD risk assessment, rather than just measuring the cholesterol carried in the lipoproteins. Since clinical features and classical risk factors are similar to other reports [8,9,17], we believe that our results might be representative and translatable to clinical practice.

HDL benefits on cardiovascular protection are mainly conferred by its capacity to promote cholesterol efflux, preventing and stabilizing atherosclerotic lesions [14,20,21,22] and its potential to protect LDL from oxidative damage [23]. The results of our study expand the understanding on the effects of HDL and LDL in ACS. A previous analysis from our institution revealed that a low level of HDL-C was the variable more closely related to being admitted for an ACS than a non-ACS [5]. In this new study, we were able to demonstrate that HDL particles were clearly dysfunctional, especially in patients admitted for STEMI, expanding the knowledge in this controversial field.

The results of our study showed that the HDL anti-atherogenic functional capacities were impaired in ACS patients, especially in the STEMI group with higher cardiac damage (elevated troponins). Interestingly, the degree of HDL-C oxidation was inversely correlated with the CEC in the reference and non-ACS patients. Oxidative stress and inflammation may occur in ACS patients and is capable of inducing pro-atherogenic modifications in lipoproteins, switching them into a dysfunctional state [24]. The degree of HDL-C oxidation was inversely correlated with the CEC in the reference healthy volunteers and non-ACS patients but not in the ACS patients because HDLs were already modified in these patients at baseline. This fact is especially evident in the STEMI patients. Hence, these observations suggest that HDL particles from subjects at the highest risk of an ACS may already have modifications in the circulation altering their functionality that are not modifiable by inducing in vitro oxidation. Nevertheless, whereas LDL-C levels were lower in patients than in the reference group, all patients had LDL particles with increased susceptibility to oxidation and impaired HDL antioxidant capacity. Moreover, cholesterol efflux capacity was significantly diminished only in ACS patients. In fact, HDL particles in ACS patients (especially the STEMI) were enlarged, probably depicting a shift into a dysfunctional state given that they are the small HDL-P ones linked to an increased cholesterol efflux and antioxidant capacities [25,26,27]. Therefore, not only the levels but also the functions of lipoproteins have a clear high impact on their contribution to ACS onset and presentation.

Some studies have suggested, based on the observation that individuals with higher levels of large HDL particles have a lower risk of CVD, that larger HDL particles are more protective against CVD than smaller HDL particles [28,29]. On the contrary, other studies showed that small, dense HDL particles may actually be more protective against CVD than larger particles [30,31,32]. This controversy suggests that the relationship between HDL particle size and CVD risk is complex and may depend on other factors such as the presence of other lipid abnormalities or genetic factors [33]. Another issue is the difficulty in accurately measuring HDL particle size. Different methods can yield different results, and there are currently no standardized methods for measuring HDL particle size [34]. Hence, though we observed that ACS patients with dysfunctional lipoproteins have larger HDL-P, more research is needed to fully understand this relationship.

Clinical registries are concordant in the findings that patients with HDL-C > 40 mg/dL have a lower incidence of cardiovascular events [5,8]. Nonetheless, the therapies that were designed to increase HDL-C levels, such as Cholesteryl Ester Transfer Protein (CETP) inhibitors [35] or nicotinic acid [36], did not reduce the incidence of major cardiovascular events. Thereafter, HDL functionality is probably impaired by some pharmacologic strategies in what reflects one of the many gaps in the knowledge of HDL particles. Our results, show significant differences in HDL particle functionality that might warrant future investigations to improve HDL functionality in subjects with high cardiovascular risk.

Our study has the limitations of being cross-sectional, performed in a single centre and having a small sample size and a low number of women. Moreover, due to logistic factors that cannot be controlled in the clinical practice, samples were obtained after admission at different times. However, despite the described limitations and the heterogeneity of groups, the study is based on a well-characterized real-world cohort of patients admitted for first-onset chest pain with on-target LDL-C levels.

In conclusion, patients treated as per guidelines in their primary care management with intermediate CVD risk that suffered a first chest pain episode had an impaired lipoprotein function, which might lead to a higher oxidative status, and an altered number/size of lipoprotein particles irrespective of the LDL-C level and optimal treatment. Interestingly, triglycerides transported by all lipoproteins were within the normal range as well as non-HDL-C levels. This study shows the relevance of changes in lipoprotein functionality and in particle number/size on first onset ACS presentation. The on-going follow-up of this cohort might add more information about recurrent events and long-term mortality according to the determinations obtained at baseline.

## 4. Materials and Methods

### 4.1. Clinical Diagnosis of Chest Pain Categories

Ninety-seven patients admitted from January 2018 to April 2018 into “Hospital San Juan de Alicante” with chest pain were clinically diagnosed as ACS (ACS; *n* = 70) or non-ACS patients (non-ACS; *n* = 27) with high cardiovascular risk. Patients were classified as ACS or non-ACS after all diagnostic tests were performed, including an exercise test, echocardiogram or angiography. In addition, ACS patients were further categorized by electrocardiogram pattern on admission into NSTEMI (*n* = 31) and STEMI (*n* = 39) (Table 1).

Non-ACS was diagnosed by the exclusion of acute ischemia (no troponin elevation and no dynamic or electrocardiographic changes suggestive of myocardial ischemia), inducible ischemia (conclusive stress test) or unstable or severe coronary lesions in the angiography, as previously published [5]. Demographic characteristics of the patients, risk factors for coronary artery disease (smoking, hypertension, dyslipidaemia and diabetes mellitus), medical history, laboratory data during the hospitalization, vital signs on admission, treatment and diagnosis at discharge were collected from all patients. A history of heart failure was codified if patients had at least one hospitalization with such diagnosis at discharge or the typical signs and symptoms of heart failure and a compatible echocardiogram. Patients underwent an echocardiography within 48 h of admission, and the left ventricular ejection fraction (LVEF) was calculated using Simpson’s method [37]. Patients were excluded from the study if they had age > 85, previous history of ischemic heart disease or heart failure, diagnosis of hypo- or hyperthyroidism, presence of previous valve disease, initial haemoglobin < 10 g/dL, initial presentation of ACS as cardiogenic shock, treatment with anti-retrovirals, pregnancy or died in the first <24 h or before the first blood test after a fasting night could be obtained (see Flowchart in Figure 3).

All patients had moderate cardiovascular disease (CVD) risk following the European Heart Score (below 5% and higher than 1%) [38] and the Framingham Risk Score (10–19%) [39] (Table 1).

Additionally, a reference group of 31 healthy, non-treated, overweight or obese volunteers without additional risk factors or clinical symptoms of disease was included for baseline comparative purposes of the novel techniques investigated in this study. Patient at admission and volunteer characteristics are shown in Appendix A.

The study complies with the Declaration of Helsinki and was approved by the Ethics Committee of Clinical Research of “Hospital San Juan de Alicante”, Spain (Ref 17/314; 7 June 2017); informed consent was obtained from all subjects.

### 4.2. Biochemical and Laboratory Parameters

Blood samples were obtained within a mean ± SEM of 2.60 ± 2.02 days. Briefly, blood samples were collected without anticoagulant or in EDTA-containing vacutainer tubes for serum and plasma preparation. Routine standard biochemical determinations including troponins and haemogram were performed for the on-going ACS registry of our institution [5]. Aliquots of both serum and plasma were kept at −80 °C for the specific assays involved in this study.

### 4.3. LDL and HDL Sample Preparation and Purity Control

LDL (density 1.019 to 1.063 g/mL) and HDL (density range 1.063–1.210 g/mL) were obtained from 1 mL plasma-EDTA from individual samples by sequential ultracentrifugation according to the method originally described by Havel et al. [40] and modified by De Juan-Franco et al. [41]. To avoid lipoperoxidation, all solutions contained 1 mmol/L EDTA and 2 μmol/L butylated hydroxytoluene (BHT) and centrifugations were performed at 4 °C using rotors stored in a cold room. Briefly, plasma was adjusted to a density of 1.019 g/mL with a concentrated salt solution (potassium bromide) and centrifuged at 225,000× *g* for 18 h in a Beckman L-60 preparative ultracentrifuge with a fixed-angle type 50.4 Ti rotor (Beckman, Brea, CA, USA). After removal of the top layer containing very low and intermediate density lipoproteins (VLDL and IDL), the density of the infranatant was adjusted to 1.063 g/mL, followed by centrifugation for 20 h at 225,000× *g* before LDL was collected from the top of the tube. Lastly, the process was repeated adjusting the plasma density to 1.210 g/mL and samples were ultracentrifuged at 225,000× *g* for 24 h at 4 °C to allow HDL to float and separate from lipoprotein-deficient serum.

In addition, LDL to be used in the TRAP assay was isolated from a pool of plasma (180 mL) obtained from normolipidemic subjects and obtained as described above in a Beckman Optima L-100 XP with a fixed-angle type 50.2 Ti (Beckman, Brea, CA, USA).

LDL and HDL fractions were dialyzed against phosphate-buffered saline (PBS) for 24 h. After dialysis, LDL and HDL protein content was determined by the colorimetric BCA assay (Pierce) and adjusted to 100 µg/mL with PBS. Samples were left protected from light at 4 °C until analysis. LDL and HDL purity was routinely analysed by electrophoresis (2 µL sample) in agarose gels using a commercial assay (SAS-MX Lipo 10 kit, Helena Biosciences, London, UK) as described by the providers.

### 4.4. Conjugated Diene Assay

Susceptibility of LDL to copper-induced oxidation was assessed by determining the formation of conjugated dienes. Briefly, freshly prepared LDL samples adjusted to 100 µg/mL with PBS were analysed in 96 well plates by incubation with a copper (II) sulphate (CuSO_4_•5H_2_O) solution at a final concentration of 5 µM. The change of absorbance was monitored for 2 h 30 min at 37 °C using a SpectraMax 190 Microplate reader (Molecular Devices, San José, CA, USA) by continuously following the formation of conjugated diene, a product of lipid peroxidation with absorbance peak at 234 nm. The total amount of conjugated diene was calculated using the molar extinction coefficient of 29,500 M^−1^cm^−1^ [42].

### 4.5. HDL Antioxidant Potential

The antioxidant potential of HDL was assessed by performing the total-radical-trapping antioxidative potential (TRAP) test [43]. This method is based on the capability of HDL to prevent LDL (control LDL) oxidation. Briefly, HDL and LDL lipoproteins were diluted in PBS 1× to a final concentration of 100 µg protein/mL. HDL from each individual subject was incubated for 4 h at 37 °C with copper (II) sulphate (CuSO_4_•5H_2_O) at final concentration of 20 µM either alone or in the presence of LDL control (plasma pool). As a baseline value, the LDL sample was incubated alone with and without CuSO_4_ during the same time period. Oxidation was stopped by adding 50 µL of EDTA 1 mM. Thereupon, 100 µL of each sample was transferred to a fluorescence 96-well plate (Corning^®^, TC Black plate with clear bottom, New York, NY, USA). Eventually, samples were incubated with 50 µL of freshly prepared DCFH-DA (2′,7′-dichlorodihydrofluorescein diacetate, Molecular Probes, Eugene, OR, USA) at a final concentration of 10 µM for 1 h 30 min at 37 °C and 100 rpm. Dichlorofluorescin-diacetate was employed as the marker of the oxidative reaction [44]. Lipid oxidation products convert DCFH to DCF, which produces intense fluorescence. The intensity of fluorescence was determined with a Typhoon FLA9500 set at λex = 500 nm and λem = 520 nm. Final fluorescence measurements were expressed as the percentage of inhibition of oxidized LDL in the presence of HDL relative to the oxidation level when LDL was incubated in absence of HDL.

### 4.6. HDL Cholesterol Efflux Capacity Assay

The cholesterol efflux capacity (CEC) of HDL was determined in cholesterol-loaded murine macrophages as previously reported [45]. To this end, J774A.1 mouse macrophages (at passage seven) were cultured in RPMI 1640 (Roswell Park Memorial Institute medium) containing 10% of heat-inactivated FBS (Foetal bovine serum), 2 mM glutamine, 100 U/mL penicillin, 100 U/mL streptomycin and 10 µg/mL gentamicin at 37 °C in a humidified atmosphere of 5% CO_2_. For the experiments, macrophages (1.5 × 10^5^ cells/well) were seeded in 6-well culture plates (Falcon 6-well Clear Flat Bottom TC-treated culture plate, Corning^®^, New York, NY, USA) and labelled for 48 h with [1α, 2α (n)-^3^H]-cholesterol] (GE Healthcare, Chicago, IL, USA) at 1 µCi per well. Cells were equilibrated overnight in 0.2% bovine serum albumin and thereafter incubated with RPMI media containing 5% ApoB-depleted serum (4 h, 37 °C) to promote cholesterol efflux from the [^3^H] cholesterol-labelled cells. ApoB-depleted serum was obtained by precipitation of ApoB particles with a solution containing phosphotungstic acid (0.484 mM) and MgCl_2_ (22 mM). ApoA-I and ApoB measurements in ApoB-depleted serum samples were determined by immunoturbidimetric assays using commercial kits adapted to a COBAS 501c autoanalyzer (Roche Diagnostics, Basilea, Switzerland). ApoB reported values in ApoB-depleted serum were below 0.06 mg/mL.

The radioactivity signal was quantified in both media and cells and the percentages of cholesterol efflux calculated by expressing the radioactive cholesterol released to the medium as the fraction (%) of the total radioactive cholesterol present in the well (radioactivity in the cell + radioactivity in medium).

### 4.7. Lipoprotein Particle Number and Size Measurements

Lipoprotein size was directly measured in serum (500 µL) by nuclear magnetic resonance (NMR) as described by Mallol et al. [46] using the two-dimensional diffusion-ordered 1H-NMR spectroscopy (2D DOSY) Liposcale^®^ (Biosfer Teslab, Reus, Spain). Briefly, particle concentration was obtained from the measured amplitudes and attenuation of their spectroscopically distinct lipid methyl group NMR signals using the 2D diffusion-ordered 1H NMR spectroscopy (DSTE) pulse. The methyl signal was surface fitted with nine Lorentzian functions associated with each lipoprotein subtype of the LDL: large, medium and small. The area of each Lorentzian function was related to the lipid concentration of each lipoprotein subtype, and the size of each subtype was calculated from their diffusion coefficient. The particle numbers for each lipoprotein subtype were calculated by dividing the lipid volume by the particle volume of a given class. The lipid volumes were determined by using common conversion factors to convert concentration units into volume units [47].

### 4.8. Statistical Analysis

Statistical analyses were conducted using StatView 5.0.1 software (SAS Institute, Cary, NC, USA) and SPSS software (IBM SPSS Statistics 25.0.0, New York, NY, USA) except when indicated. Data are expressed by the number of cases (qualitative variable) and as mean ± standard error of the mean (SEM) or median [IQR] for the quantitative variable. The normal distribution of variables was analysed by the Kolmogorov–Smirnov test. Differences between characteristics of the groups were analysed by unpaired Student’s *t*-test or an analysis of variance (ANOVA) for parametric variables. A Bonferroni post hoc test was run for two group comparisons after ANOVA. Slope differences between groups in regression analysis were assessed by analysis of covariance (ANCOVA). When normality failed, Mann–Whitney or Wilcoxon tests was performed for non-parametric variables. When needed, chi-squared analysis was performed as indicated in the Results section. All reported *p*-values are two-sided, and a *p*-value of 0.05 or less was considered to indicate statistical significance.

## Figures and Tables

**Figure 1 ijms-24-05391-f001:**
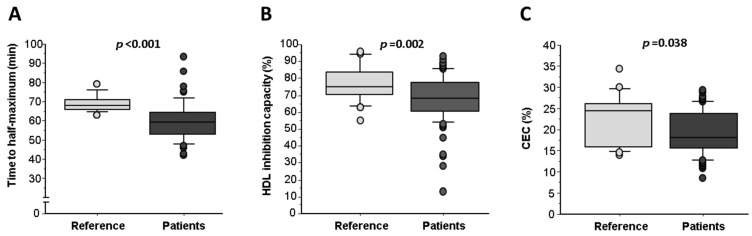
Lipoprotein functionality in the reference group (*n* = 31) vs. patients (*n* = 97). (**A**) Time to half-maximum oxidation (minutes). (**B**) HDL inhibition of oxidation (%) and (**C**) cholesterol efflux capacity (CEC [%]). Results are expressed as median and quartile 1–3 range (horizontal lines). Student’s *t*-test values of *p* < 0.05 for independent samples are considered significant.

**Figure 2 ijms-24-05391-f002:**
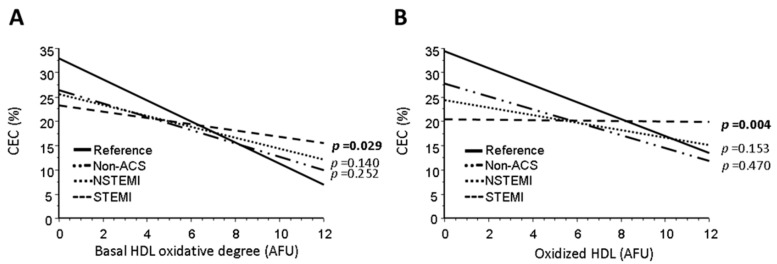
Bivariate regressions. (**A**) Basal HDL oxidative degree (AFU) versus cholesterol efflux capacity (CEC [%]) and (**B**) copper-oxidized HDL (AFU) vs. CEC [%]. Results for the reference individuals, Non-ACS, NSTEMI and STEMI patients. Slope differences of groups versus the reference group (healthy) were assessed by ANCOVA analysis. Values of *p* < 0.05 are considered significant. Abbreviations: ACS = Acute Coronary Syndrome, CEC = Cholesterol Efflux Capacity, NSTEMI = Non-ST-Segment-elevation myocardial infarction, STEMI = ST-Segment-Elevation Myocardial Infarction, HDL = High Density Lipoprotein.

**Figure 3 ijms-24-05391-f003:**
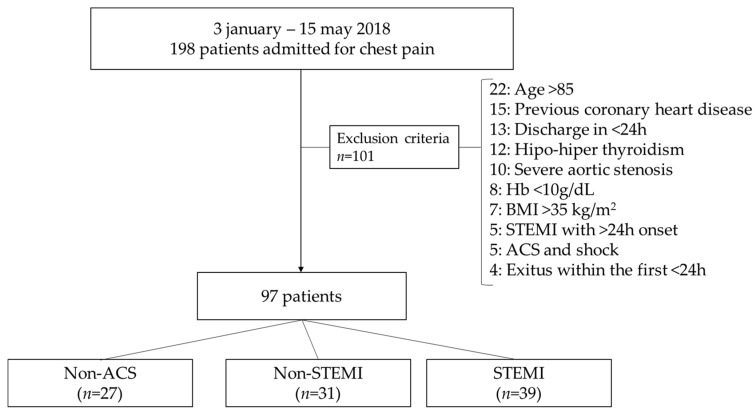
Flowchart of patient selection and exclusion criteria. Exclusion criteria includes the number of patients excluded for each criteria.

**Table 1 ijms-24-05391-t001:** Patient characteristics: Electrocardiogram classification. The chi-squared test was applied for categorical variables and the ANOVA test for numerical variables. The Bonferroni post hoc test was run for two group comparisons after ANOVA: *****, Significant differences versus non-ACS; **‡**, Significant differences between STEMI and NSTEMI. Values of *p* < 0.05 are considered significant. Abbreviations: ACS = acute coronary syndrome, ApoA-I = Apolipoprotein A-I, NSTEMI = non-ST-segment-elevation myocardial infarction, STEMI = ST-segment-elevation myocardial infarction, BMI = body mass index, COPD = chronic obstructive pulmonary disease, BNP = B-type natriuretic peptide, LVEF = left ventricular ejection fraction. GOT = glutamic-oxaloacetic transaminase, GPT = glutamic-pyruvic transaminase, ACE inhibitor = angiotensin-converting enzyme inhibitor, OAD = oral anti-diabetic agents, DPP-4 inhibitors = dipeptidyl peptidase 4 inhibitors.

		ACS (*n* = 70)		*p*-Value
	Non-ACS (*n* = 27)	NSTEMI (*n* = 31)	STEMI (*n* = 39)	
Age (years, mean ± SEM)	65.8 ± 2.6	70.3 ± 1.9	60.0 ± 1.7 ***‡**	**0.001**
Male/Female, *n*	14/13	26/5	32/7	**0.007**
Weight (Kg)	84.1 ± 5.1	79.5 ± 2.7	85.2 ± 2.5	0.375
BMI (Kg/m^2^, mean ± SEM)	29.6 ± 1.4	27.7 ± 0.6	28.7 ± 0.7	0.354
**Risk Factor, *n* (%)**				
Diabetes Mellitus	8 (30%)	10 (32%)	6 (15%)	0.210
Hypertension	15 (55%)	25 (81%)	15 (38%)	**0.002**
Obesity (>30%)	8 (30%)	7 (23%)	13 (33%)	0.517
Dyslipidaemia	11 (41%)	17 (55%)	11 (28%)	0.078
Smoking habits	4 (15%)	5 (16%)	22 (56%)	**<0.001**
COPD	0 (0%)	3 (10%)	2 (5%)	0.251
**Lipid profile**				
Total cholesterol (mg/dL)	182.2 ± 8.6	167.5 ± 6.6	169.1 ± 5.6	0.292
Triglycerides (mg/dL)	140.3 ± 11.1	154.2 ± 21.4	143.6 ± 11.7	0.811
LDL-C (mg/dL)	102.4 ± 8.4	95.8 ± 5.4	101.8 ± 5.2	0.715
HDL-C (mg/dL)	52.0 ± 2.8	44.3 ± 2.4 *****	39.4 ± 1.7 *****	**<0.001**
ApoA-I (mg/ml)	1.8 ± 0.1	1.5 ± 0.1 *****	1.4 ± 0.1 *****	**<0.001**
HDL-C/LDL-C	0.61 ± 0.07	0.50 ± 0.04	0.43 ± 0.03 *****	**0.021**
Triglycerides/HDL-C	3.1 ± 0.3	4.2 ± 0.8	4.1 ± 0.5	0.395
Non-HDL-C (mg/dL)	130.2 ± 8.8	123.1 ± 6.7	129.6 ± 5.7	0.734
Lp(a) (mg/dL)	27.3 ± 5.9	35.1 ± 6.1	49.7 ± 7.3 *****	0.058
**Cardiac parameters**				
BNP (pg/mL)	401 ± 220	1600 ± 556	2082 ± 594 *****	0.081
Troponin-I (ng/L)	720 ± 484	9355 ± 3208	83,307 ± 17,824 ***‡**	**<0.001**
LVEF (%)	59.4 ± 1.5	57.2 ± 1.5	51.7 ± 1.9 ***‡**	**0.007**
**Hepatic and renal parameters**				
GOT (UI/l)	24.2 ± 2.8	40.2 ± 8.1	66.2 ± 11.4 ***‡**	**0.006**
GPT (UI/l)	24.7 ± 3.9	31.6 ± 5.1	40.1 ± 4.0 *****	**0.050**
Creatinine (mg/dL)	0.8 ± 0.06	1.0 ± 0.05 *****	0.9 ± 0.04	0.060
**Medication at admission, *n* (%)**				
Acetylsalicylic acid	7 (26%)	7 (23%)	2 (5%)	**0.044**
Clopidogrel	2 (7%)	0 (0%)	1 (3%)	0.259
Angiotensin II receptor blocker	12 (44%)	14 (45%)	8 (20%)	**0.048**
ACE inhibitor	2 (7%)	4 (13%)	2 (5%)	0.493
Diuretics	7 (26%)	6 (19%)	4 (10%)	0.245
Calcium antagonists	2 (7%)	8 (26%)	2 (5%)	**0.022**
Beta-Blockers	8 (30%)	9 (29%)	1 (3%)	**0.004**
Omeprazole	6 (22%)	11 (35%)	3 (8%)	**0.016**
Statins	8 (30%)	11 (35%)	9 (23%)	0.521
Fibrates	1 (4%)	1 (3%)	2 (5%)	0.916
OADs	4 (15%)	7 (23%)	3 (8%)	0.212
Metformin	5 (16%)	4 (13%)	2 (5%)	0.228
DPP-4 inhibitors	2 (7%)	2 (6%)	1 (3%)	0.631
**Others, *n* (%)**				
Coronary stent	0 (0%)	7 (23%)	12 (31%)	**0.007**
Cathetherization/Revascularization	8 (30%)	15 (48%)	32 (82%)	**<0.001**
**Risk Scores (%)**				
Framingham risk score	10.4 ± 1.4	17.9 ± 1.2	12.1 ± 1.1	**<0.001**
European Heart Score	2.7 ± 0.3	4.6 ± 0.3	3.1 ± 0.3	**<0.001**

**Table 2 ijms-24-05391-t002:** Lipoprotein particle number and size. Results are expressed as mean ± SEM. The ANOVA test was applied for multiple comparisons; the Bonferroni post hoc test was run for two group comparisons after ANOVA: *****, significant changes versus non-ACS group; **‡**, significant changes between NSTEMI and STEMI groups. Values of *p* < 0.05 are considered significant. Abbreviations: ACS = acute coronary syndrome, NSTEMI = non-ST-segment-elevation myocardial infarction, STEMI = ST-segment-elevation myocardial infarction, VLDL = very low density lipoprotein, IDL = intermediate density lipoprotein, HDL = high density lipoprotein, TG = triglycerides.

		ACS (*n* = 69)		*p*-Value
	Non-ACS (*n* = 25)	NSTEMI (*n* = 30)	STEMI (*n* = 39)	
**Triglyceride content in lipoprotein particles (mg/dL)**				
VLDL-TG	85.6 ± 6.9	77.2 ± 5.6	87.1 ± 6.5	0.504
IDL-TG	11.4 ± 0.3	11.2 ± 0.5	12.1 ± 0.6	0.428
LDL-TG	14.1 ± 0.9	13.4 ± 0.7	15.0 ± 0.7	0.312
HDL-TG	13.9 ± 0.5	12.2 ± 0.5	12.3 ± 0.7	0.148
**VLDL particle number (nmol/L)**				
Small VLDL-P	54.7 ± 4.7	48.9 ± 3.8	54.8 ± 4.3	0.549
Medium VLDL-P	5.7 ± 0.4	5.6 ± 0.3	6.8 ± 0.4 **‡**	0.066
Large VLDL-P	1.5 ± 0.1	1.4 ± 0.1	1.5 ± 0.1	0.488
Total VLDL-P	61.9 ± 5.1	55.9 ± 4.2	63.1 ± 4.7	0.513
**LDL particle number (nmol/L)**				
Small LDL-P	691.9 ± 28.6	642.8 ± 19.6	677.5 ± 24.7	0.387
Medium LDL-P	367.3 ± 32.4	312.8 ± 17.1	336.7 ± 14.9	0.225
Large LDL-P	194.3 ± 9.9	184.9 ± 4.8	191.3 ± 4.9	0.594
Total LDL-P	1253.5 ± 62.7	1140.5 ± 35.2	1205.5 ± 40.6	0.257
**HDL particle number (nmol/L)**				
Small HDL-P	18.3 ± 0.8	14.8 ± 0.7 *****	13.8 ± 0.6 *****	**<0.001**
Medium HDL-P	10.6 ± 0.4	10.7 ± 0.3	10.0 ± 0.2	0.171
Large HDL-P	0.31 ± 0.01	0.30 ± 0.01	0.30 ± 0.01	0.501
Total HDL-P	29.2 ± 1.0	25.8 ± 0.8 *****	24.1 ± 0.6 *****	**<0.001**
**Average particle diameter (nm)**				
VLDL	42.1 ± 0.05	42.1 ± 0.04	42.2 ± 0.04 *****	0.119
LDL	21.0 ± 0.06	21.0 ± 0.04	21.0 ± 0.04	0.913
HDL	8.3 ± 0.02	8.4 ± 0.02 *****	8.4 ± 0.02 *****	**0.004**

**Table 3 ijms-24-05391-t003:** Differences between the reference and non-ACS/ACS groups. Results are expressed as mean ± SEM. The ANOVA test was applied for multiple comparisons and the Bonferroni post hoc test was run for two group comparisons after ANOVA: *****, significant changes versus the reference group; **‡**, significant changes between Non-ACS and ACS groups. Values of *p* < 0.05 are considered significant. Abbreviations: ACS = Acute Coronary Syndrome, TRAP = The Total-Radical-Trapping Antioxidative Potential, CD max = Maximum of Conjugated Dienes, V max = Maximum Velocity, BNP = B-Type Natriuretic Peptide, LVEF = Left Ventricular Ejection Fraction.

		Patients (*n* = 97)		*p*-Value
	Reference Group (*n* = 31)	Non-ACS (*n* = 27)	ACS (n = 70)	
**HDL functionality**				
TRAP (% HDL inhibition capacity)	76.9 ± 1.9	66.3 ± 3.2 *****	68.9 ± 1.5 *****	**0.005**
Cholesterol Efflux (% CE)	22.3 ± 1.4	19.7 ± 1.0	19.5 ± 0.6 *****	0.116
HDL baseline oxidation (RFU)	4.8 ± 0.3	5.0 ± 0.4	5.2 ± 0.2	0.671
**LDL susceptibility to oxidation**				
CD max (nmol CD/mg LDL)	363 ± 4.1	369 ± 13.0	345 ± 6.7 **‡**	0.071
V max (CD/min/mg LDL)	4.3 ± 0.1	5.5 ± 0.6 *****	4.7 ± 0.3	0.095
Time to half-maximum (min)	69.3 ± 1.0	61.0 ± 2.3 *****	59.3 ± 1.7 *****	**<0.001**

## Data Availability

The data presented in this study are available on request, due to privacy restrictions.

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
