# Peer review of "HDL Function and Size in Patients with On-Target LDL Plasma Levels and a First-Onset ACS"

_ijms, 2023, doi:10.3390/ijms24065391_

Round 1

Reviewer 1 Report

Major comments

 1.         Tables 1 and 2 studied patient characteristics and lipoprotein particle numbers of Non-ACS and ACS (NSTEMI and STEMI). Table 3 compares the differences between the reference and patients (Non-ACS and ACS). This reviewer recommends reworking these Tables for clarity. For example, Table 3 might be better with HDL functionality and LDL susceptibility to oxidation, excluding Lipid profile and Cardiac parameters.

2.  It is generally known that high LDL-C is associated with a high risk of cardiovascular events.  However, in this study by Alberto Cordero et al., LDL-C is lower in patients when comparing patients and references (Table 3). Why is this?

3. It is important to state in the explanation of Table 1 that there was no significant difference in LDL-C and Non-HDL-C levels between the Non-ACS and ACS (NSTEMI and STEMI) groups.

4. Many studies have reported low HDL-C and Apo-AI levels in patients. In this study by Alberto Cordero et al., Table 1 showed that HDL-C and Apo-AI levels were lower in ACS patients than in non-ACS. This study was carried out after separating patients into non-ACS and ACS (NSTEMI and STEMI). Is this new?

5. Previous studies indicated that the reduced concentrations of circulating HDL particles could be superior to HDL cholesterol. So, measurements of HDL particle numbers bear a potential for the improved assessment of cardiovascular risk. In the authors’ study, table 2 indicated that HDL particle number was lower in ACS. It is not a particularly new finding.

On the other hand, there are various reports on the relationship between HDL size and myocardial risk. For example, some studies reported that the plasma concentrations of specific HDL subpopulations could be hypothesized to reveal stronger associations with cardiovascular risk. NMR measurements allow distinguishing between large (size 9.4–14.0 nm), medium (8.3–9.3 nm), and small (7.3–8.2 nm) HDL particles. Among these HDL subpopulations, levels of large HDL frequently display inverse relationships with cardiovascular risk. In contrast, concentrations of small HDL particles typically reveal positive correlations with the risk. So, discussing no match results between the authors’ study and previous data is recommended.

6. It has been reported that a decrease in HDL function, especially cholesterol efflux capacity, increases the risk of atherosclerosis and myocardial infarction. Measuring CEC is said to be suitable for determining HDL function. In this study, the authors also showed decreased CEC in patients. What are the new findings?

7. What is the N of the Reference and Patients in the experiment in Figure 1, N=31 and N=97?

8  Is Healthy the same as Reference in Figure 2?

9. The underlined part below is not easy to understand.

The oxidative condition of HDL was determined by fluorometry, and it was inversely associated with the CEC in healthy volunteers (Figure 2). Interestingly, HDL function was similar in non-ACS patients and in the healthy reference group (Figure 2A and Figure S1). However, STEMI patients had significantly lower slope and Y-intercept value compared to the reference group (p=0.029 and p=0.012, respectively) after inducing HDL-copper mediated oxidation (Figure 2B and Figure S2), (p=0.003 and p<0.001 respectively).

10. In Figure 2, oxidation in the healthy group causes a decrease in CEC, indicating that oxidation causes HDL dysfunction. However, why in ACS, CEC function is low, and the effect of oxidation on CEC is small?

11. The following English sentences in the Discussion section (lines 327-331) are difficult to understand.

In conclusion, patients treated as per guidelines in their primary care management, with intermediate CVD risk that suffered a first chest pain episode had an impaired lipoprotein function with a higher oxidative status (both in HDL-C and LDL-C) and an altered number/size of lipoprotein particles irrespective of the LDL particles irrespective of the LDL-C level and optimal treatment.

12.  Text in lines 327-331 mentions a higher oxidative status (both in HDL-C and LDL-C) in patients. Based on which of the experiments in this study did the authors mention that HDL-C and LDL-C are both in higher oxidative status?

Minor comments

1.       LDL and HDL levels in the Abstract should be changed to LDL-C and non-HDL-C levels.

Author Response

REVIEWER 1

We thank the reviewer 1 for the detailed review of our manuscript. We consider that the reviewer’s comments have been very helpful and have substantially improved the quality and clarity of our study.

Comments and Suggestions for Authors

Major comments

  1. Tables 1 and 2 studied patient characteristics and lipoprotein particle numbers of Non-ACS and ACS (NSTEMI and STEMI). Table 3 compares the differences between the reference and patients (Non-ACS and ACS). This reviewer recommends reworking these Tables for clarity. For example, Table 3 might be better with HDL functionality and LDL susceptibility to oxidation, excluding Lipid profile and Cardiac parameters.

Following the reviewer’s advice, we have now excluded lipid profile and cardiac parameters from Table 3.

  1. It is generally known that high LDL-C is associated with a high risk of cardiovascular events. However, in this study by Alberto Cordero et al., LDL-C is lower in patients when comparing patients and references (Table 3). Why is this?

Indeed, the objective of the study was to include patients with first episode of chest pain and target levels of LDL in order to identify other parameters beyond plasma lipid levels that could be modified in these patients and that could have triggered the CVD event. To improve the clarity and message of our study we have change the title to: “HDL function and size in patients with on-target LDL levels and a first onset ACS”

Following the reviewer’s advice (see comment 1), we have now excluded lipid profile and cardiac parameters from Table 3.  Nevertheless, this can be observed in Table S1, where the reference group has significantly higher LDL-C compared to Patients group. Reasons are the following: a) as it was mentioned in the manuscript, the patient group were patients treated as per guidelines, with very well controlled lipid levels (on-target levels for primary prevention); and b) the reference group are volunteers that are non-treated individuals considered “healthy” with a little overweight.

As the reviewer mentions, high LDL-C is associated with a high risk of cardiovascular events, but the selected study patients had a CVD event despite having LDL-C and non-HDL-C levels on target. Hence, the aim of this study has been to study other parameters beyond lipid levels that could be modified in these patients and that could have triggered an ACS.

  1. It is important to state in the explanation of Table 1 that there was no significant difference in LDL-C and Non-HDL-C levels between the Non-ACS and ACS (NSTEMI and STEMI) groups.

We added this information in blue colour in the manuscript.

  1. Many studies have reported low HDL-C and Apo-AI levels in patients. In this study by Alberto Cordero et al., Table 1 showed that HDL-C and Apo-AI levels were lower in ACS patients than in non-ACS. This study was carried out after separating patients into non-ACS and ACS (NSTEMI and STEMI). Is this new?

See responses to 4 and 5 together below 5.

  1. Previous studies indicated that the reduced concentrations of circulating HDL particles could be superior to HDL cholesterol. So, measurements of HDL particle numbers bear a potential for the improved assessment of cardiovascular risk. In the authors’ study, table 2 indicated that HDL particle number was lower in ACS. It is not a particularly new finding.

On the other hand, there are various reports on the relationship between HDL size and myocardial risk. For example, some studies reported that the plasma concentrations of specific HDL subpopulations could be hypothesized to reveal stronger associations with cardiovascular risk. NMR measurements allow distinguishing between large (size 9.4–14.0 nm), medium (8.3–9.3 nm), and small (7.3–8.2 nm) HDL particles. Among these HDL subpopulations, levels of large HDL frequently display inverse relationships with cardiovascular risk. In contrast, concentrations of small HDL particles typically reveal positive correlations with the risk. So, discussing no match results between the authors’ study and previous data is recommended.

We reply to both answer 4 and 5 together.

Regarding HDL-C and Apo-AI levels, it is not new that patients have lower levels of both HDL-C and Apo-AI, but we consider that this information needs to be provided along with the other functionality parameters in order to fully understand the possible underlying causes of the CVD event in this specific subset of patients that have a first ACS even though they have LDL levels of 100mg/dl. Regarding HDL particle number, our aim is to show that these ACS patients, which have shown impaired lipoprotein functionality, have a decreased proportion of small HDL-P and increased proportion of large HDL-P. These findings are supported by previous studies:

  • Small-HDL-P is the best acceptor of ABCA1-mediated cholesterol efflux from macrophages, and Small and dense HDL particles also protect LDL from oxidation by removing phospholipid hydroperoxides from LDL (Duparc T et al. 2020, Kontush A et al. 2006 and 2010).
  • Tanaka S et al. 2019, showed that during an acute phase of septic shock there was a shift towards large HDL that the authors associated to a major dysfunction of these lipoproteins.
  • Ortiz-Muñoz G. et al. 2016, showed the same trend in patients with acute cerebral infarction as they had an increased proportion of large HDL particles.
  • Recently, Tang X et al. 2021, showed that ACS patients with high hsCRP levels had HDL particles that shifted to a larger size, a phenomenon that was accompanied by decreased CEC. 
  • Lastly, The IDEAL and EPIC-Norfolk Studies (2008) observed that very large HDL particles are associated with increased CAD risk hypothesising that very large HDLs, which are cholesterol enriched, may at some point become cholesterol donors instead of acceptors.

Nevertheless, as the reviewer is stating, there are other studies suggesting the opposite. We added a paragraph (blue colour) in the discussion mentioning the diverse and controversial data existing in the literature.

  1. It has been reported that a decrease in HDL function, especially cholesterol efflux capacity, increases the risk of atherosclerosis and myocardial infarction. Measuring CEC is said to be suitable for determining HDL function. In this study, the authors also showed decreased CEC in patients. What are the new findings?

The reviewer has asked many times about novelty of our findings. Please refer to response to question 2. The novelty is to characterize, regarding HDL functionality and LDL oxidative properties, why these patients with well controlled risk factors for primary prevention suffer an episode of chest pain and are eventually diagnosed with ACS. This is a very novel study not done before.

 It is important to notice, that most studies reporting impaired HDL functionality in CVD populations are performed in individuals with high LDL levels, what might contribute to the progression of atherosclerotic plaques. The patient cohort of our study has interesting characteristics, such as on-target lipid levels (LDL-C and non-HDL-C) for primary prevention. Therefore, we have investigated the importance of functionality without the possible interference of elevated cholesterol levels. Current guidelines are still focused on targeting high lipid levels, and there are not therapeutic strategies aimed to improve lipoprotein functionality. These findings might contribute to highlight the relevance of improving the “quality” rather than the quantity of HDL-C, even in cohorts or patients with on-target LDL levels.

  1. What is the N of the Reference and Patients in the experiment in Figure 1, N=31 and N=97?

Thank you. This has been now mentioned in the Figure 1 footnote in the manuscript.

  1. Is Healthy the same as Reference in Figure 2?

We apologize for the misunderstanding; we have changed it for Reference to avoid confusions.

  1. The underlined part below is not easy to understand.

“The oxidative condition of HDL was determined by fluorometry, and it was inversely associated with the CEC in healthy volunteers (Figure 2). Interestingly, HDL function was similar in non-ACS patients and in the healthy reference group (Figure 2A and Figure S1). However, STEMI patients had significantly lower slope and Y-intercept value compared to the reference group (p=0.029 and p=0.012, respectively) after inducing HDL-copper mediated oxidation (Figure 2B and Figure S2), (p=0.003 and p<0.001 respectively).”

R/ Thank you for your comment. We do not find a major problem but we will modify the sentence as shown below:

“In the reference group induced-HDL oxidation (determined by fluorometry) was inversely correlated with the CEC (Figure 2), and, interestingly, a similar relationship was found in non-ACS patients (Figure 2A and Figure S1). However, ACS-STEMI patients had significantly lower slope and Y-intercept value compared to the reference group (p=0.029 and p=0.012, respectively) after inducing HDL oxidation (Figure 2B and Figure S2), (p=0.003 and p<0.001 respectively)”.

  1. In Figure 2, oxidation in the healthy group causes a decrease in CEC, indicating that oxidation causes HDL dysfunction. However, why in ACS, CEC function is low, and the effect of oxidation on CEC is small?

As it was stated in the manuscript, “the degree of HDL-C oxidation was inversely correlated with the CEC in the reference group of volunteers and non-ACS patients but not in the ACS. These observations suggest that HDL particles from subjects at the highest risk of an ACS may already have modifications in the circulation altering their functionality that are not modifiable by inducing in vitro oxidation.”

However, we will modify the text as it follows with the aim to improve its clarity:

These observations suggest that HDL particles from subjects at the highest risk of an ACS may already have modifications in the circulation altering their functionality that are not modifiable by inducing in vitro oxidation.”

Hence, it’s possible that a dysfunctional condition (higher baseline oxidation status, inflammation or other unknown) or conditions in these patients trigger modifications to HDL particles inducing a loss of function. Therefore, the effect of laboratory-induced oxidation on CEC is small because these particles seem to be already defective in the circulation of these patients.

  1. The following English sentences in the Discussion section (lines 327-331) are difficult to understand.

In conclusion, patients treated as per guidelines in their primary care management, with intermediate CVD risk that suffered a first chest pain episode had an impaired lipoprotein function with a higher oxidative status (both in HDL-C and LDL-C) and an altered number/size of lipoprotein particles irrespective of the LDL particles irrespective of the LDL-C level and optimal treatment.

We are sorry, but in the manuscript we do not find what the reviewer is stating, we did not write this part in the original text “an altered number/size of lipoprotein particles irrespective of the LDL particles irrespective of the LDL-C level and optimal treatment.”

In the original text, what is stated is the following:

In conclusion, patients treated as per guidelines in their primary care management, with intermediate CVD risk, that suffered a first chest pain episode had an impaired lipoprotein function with a higher oxidative status (both in HDL-C and LDL-C) and an altered number/size of lipoprotein particles irrespective of the LDL-C level and optimal treatment.

  1. Text in lines 327-331 mentions a higher oxidative status (both in HDL-C and LDL-C) in patients. Based on which of the experiments in this study did the authors mention that HDL-C and LDL-C are both in higher oxidative status?

 We apologize, as we might have not written this paragraph in a clear way. What we mean is that the impaired functionality of both LDL and HDL, might have led to a higher oxidative status in ACS patients. We have modified this part in the manuscript.

Minor comments

  1. LDL and HDL levels in the Abstract should be changed to LDL-C and non-HDL-C levels.

We thank the reviewer for noticing this mistake that has been amended in the original manuscript.

Reviewer 2 Report

The manuscript is well organized and presents interesting results with potential clinical relevance, improving the knowledge of the topic and, therefore, it deserves publication.

Although there are some limitations, well evidenced by the Authors, in the cohort selection criteria and in statistical analyses, they do not seem to affect the quality of the paper.

However, the text contains numerous inaccuracies which require careful revision.

Some results are not clearly presented, even if their discussion is generally clear.

-          Major points:

o   the graphical abstract is missing;

o   line 511-512: the reported supplementary materials “Supplementary Materials: The following supporting information can be downloaded at: www.mdpi.com/xxx/s1, Figure S1: title; Table S1: title; Video S1: title.” do not correspond to the uploaded files; video S1 is missing;

o   Lack of data and discussion on Lp(a) contribution to the assessed parameters in all patients, particularly in STEMI patients, which present significant higher levels of the lipoprotein with respect to reference group;

o   in Materials & Methods section, the protocols reported in paragraph 4.5. “HDL antioxidant potential” and 4.6 “HDL Cholesterol Efflux Capacity Assay”, could be further detailed;

-           

-          Minor points:

o   line 32 (abstract): ACS-STEMI acronym should be given in full.

o   same observation for CEC acronym (stated for the first time in Figure 1 legend and given in full in Materials & Methods section).

o   in Table S2 the symbol Ç‚ is not present, although it is reported in the legend;

o   in Table S3, please carefully check significance values and corresponding p-values

o   in table S2, S3 and S4 legends, it should be specified to which comparisons p-values refer to

o   in Figure S1 and S2 legends, NICP is surely a typing mistake.

Author Response

REVIEWER 2

We thank the reviewer 2 for the detailed review of our manuscript and the positive opinion of our work. We consider that the reviewer’s comments have been very helpful and have substantially improved the quality and clarity of our study.

Comments and Suggestions for Authors

The manuscript is well organized and presents interesting results with potential clinical relevance, improving the knowledge of the topic and, therefore, it deserves publication.

Although there are some limitations, well evidenced by the Authors, in the cohort selection criteria and in statistical analyses, they do not seem to affect the quality of the paper.

However, the text contains numerous inaccuracies which require careful revision.

Some results are not clearly presented, even if their discussion is generally clear.

Thank you for your very positive comments

-          Major points:

  • the graphical abstract is missing;

We apologize for this inconvenience. We uploaded the graphical abstract in the application form of IJMS as a separated file (image). We do not understand what might have happened, but we will upload the graphical abstract again in this new version of the article.

  • line 511-512: the reported supplementary materials “Supplementary Materials: The following supporting information can be downloaded at: www.mdpi.com/xxx/s1, Figure S1: title; Table S1: title; Video S1: title.” do not correspond to the uploaded files; video S1 is missing;

 The reviewer is absolutely right, we are grateful to the reviewer to have noticed this mistake that is now amended in the original manuscript (changes in the manuscript are showed in blue colour).

  • Lack of data and discussion on Lp(a) contribution to the assessed parameters in all patients, particularly in STEMI patients, which present significant higher levels of the lipoprotein with respect to reference group;

The reviewer is correct. Data on Lp(a) is presently been investigated in our laboratory. It was left in the manuscript to define the patient cohort. 

  • in Materials & Methods section, the protocols reported in paragraph 4.5. “HDL antioxidant potential” and 4.6 “HDL Cholesterol Efflux Capacity Assay”, could be further detailed;

As suggested, some details that are relevant, such as obtaining the ApoB depleted serum and the measurements of ApoA and ApoB  have been included . We detailed this information (colour blue).

-          Minor points:

  • line 32 (abstract): ACS-STEMI acronym should be given in full.

Done. Thank you.

  • same observation for CEC acronym (stated for the first time in Figure 1 legend and given in full in Materials & Methods section).

Done. Thank you.

  • in Table S2 the symbol Ç‚is not present, although it is reported in the legend

 We apologize and thank the reviewer for noticing this mistake, as in Table S2 there are only significant changes versus Non-ACS group, reported with * symbol.

  • in Table S3, please carefully check significance values and corresponding p-values

We take very seriously our statistical analysis, and we have carefully checked our statistics again, we have not detected any inaccuracies in the p-values reported. Inconsistencies have been only found in the information provided in table legends. We thank the reviewer for noticing these mistakes and improving the quality and clarity of our manuscript.  Table legends are revised and corrected.

  • in table S2, S3 and S4 legends, it should be specified to which comparisons p-values refer to

Revised and done in Table legends (blue  colour).

  • in Figure S1 and S2 legends, NICP is surely a typing mistake.

We thank the reviewer to have noticed the typo; we have corrected it.

Again, we thank the reviewer to have noticed these inaccuracies in the supplementary materials.

Round 2

Reviewer 1 Report

The response to the reviewer's comments is satisfactory.